# Concentration Distribution Learning from Label Distributions

**Jiawei Tang** [1 2]   **Yuheng Jia** [1 2]

## Abstract

Label distribution learning (LDL) is an effective method to predict the relative label description degree (a.k.a. label distribution) of a sample. However, the label distribution is not a complete representation of an instance because it overlooks the absolute intensity of each label. Specifically, it's impossible to obtain the total description degree of hidden labels that not in the label space, which leads to the loss of information and confusion in instances. To solve the above problem, we come up with a new concept named background concentration to serve as the absolute description degree term of the label distribution and introduce it into the LDL process, forming the improved paradigm of concentration distribution learning. Moreover, we propose a novel model by probabilistic methods and neural networks to learn label distributions and background concentrations from existing LDL datasets. Extensive experiments prove that the proposed approach is able to extract background concentrations from label distributions while producing more accurate prediction results than the state-of-the-art LDL methods. The code is available in https://github.com/seutjw/CDL-LD.

## 1. Introduction

Multi-label learning (MLL) (Tsoumakas & Katakis, 2007; Zhang & Zhou, 2014; Peng et al., 2025; Li et al., 2025) is a well-researched machine learning paradigm where a set of labels is assigned to each sample. In MLL, a logical value (0 or 1) is used to indicate whether a label is capable of describing a sample. But each label's relative importance to a sample cannot be accurately described by this

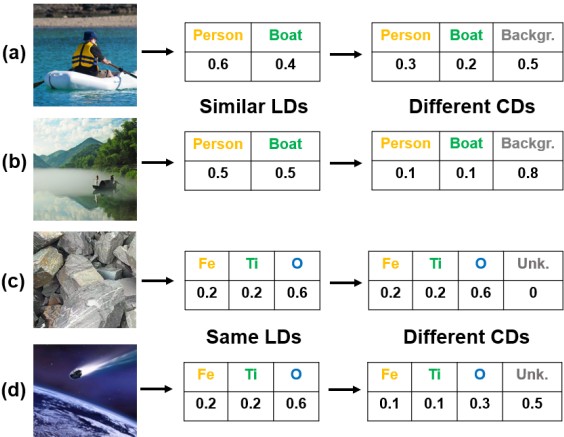

*Figure 1.* The LD-CD diagrams of two pairs (a,b and c,d) of pictures that share similar or same LDs. After introducing the background concentration, their CDs can be distinguished easily.

binary value. To this end, label distribution learning (LDL) (Geng, 2016) was developed, which uses real numbers to illustrate how important a label is to a particular sample. All labels' description degrees make up a label distribution (LD). Specifically, let $d_{\boldsymbol{x}}^l$ stand for the description degree of the label $l$ to the instance $\boldsymbol{x}$, which is governed by the non-negative constraint $d_{\boldsymbol{x}}^l \in [0, 1]$ and the sum-to-one constraint $\sum_l d_{\boldsymbol{x}}^l = 1$. Unlike traditional MLL, LDL is a more general paradigm that attempts to predict the LD for unseen samples.

In LDL, ground-truth LDs are considered complete representations of the corresponding instances. However, there exist some cases where we cannot describe an instance perfectly only with its label distribution. Fig. 1(a) and Fig. 1(b) show two pictures of boating. Although there is a significant difference between the sizes of people and boats in the two pictures, similar LDs are assigned to them due to the similar relative proportions of the two labels ("Person" and "Boat"), which goes strongly against our intuition. To solve this problem, we introduce the concept of background concentration, which represents the description degree of the complementary set of all existing labels. In this case, it can be regarded as the proportion of the background in the two pictures, which differs a lot from each other. Appending background concentration terms (Backgr. in Fig. 1(a)

[1]School of Computer Science and Engineering, Southeast University, Nanjing 210096 [2]Key Laboratory of New Generation Artificial Intelligence Technology and Its Interdisciplinary Applications (Southeast University), Ministry of Education, China. Correspondence to: Yuheng Jia <yhjia@seu.edu.cn>.

*Proceedings of the 42$^{nd}$ International Conference on Machine Learning*, Vancouver, Canada. PMLR 267, 2025. Copyright 2025 by the author(s).

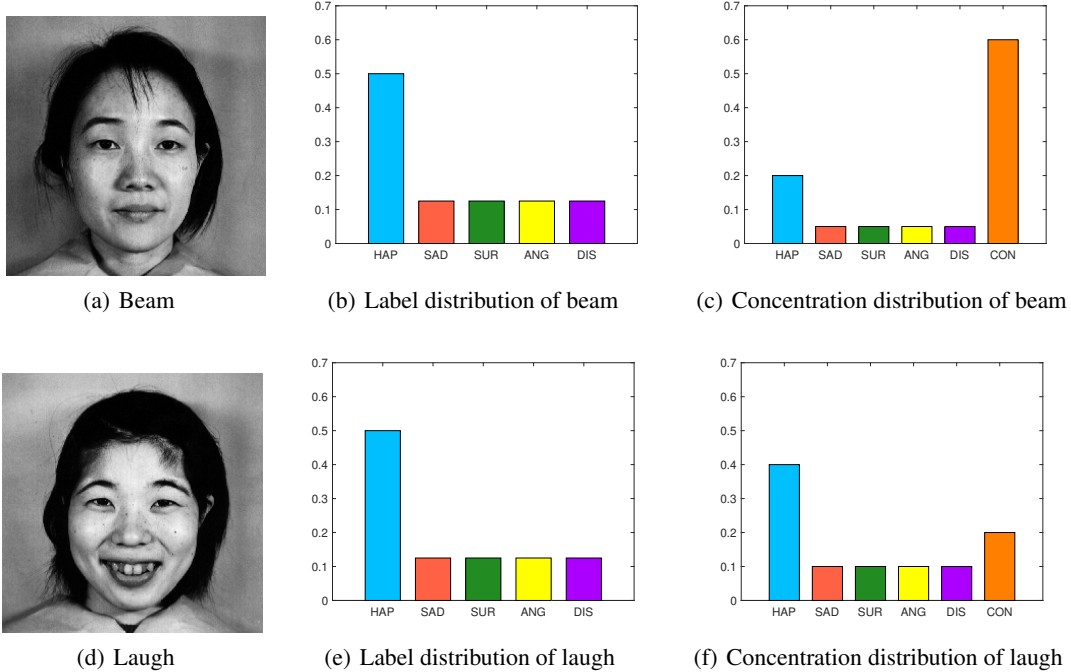

*Figure 2.* Label and concentration distributions of the images in (a) and (d). The LDs and CDs are shown in bar charts for ease of observation. By introducing the background concentration term, i.e., CON in (f) and (i), the two images that have identical label distributions can be distinguished.

and (b)) to the original LDs, we get **concentration distributions (CDs)**. CDs contain more information than LDs, helping better describe an instance. In this paper, we define concentration distribution learning (CDL) as the process of learning the background concentration $\mu$ and the label distribution $b$ simultaneously, which aims to learn concentration distribution vectors $c_d = [b, \mu]$ from concentration distribution datasets.

In fact, the concept of background concentration is very common and important in reality. For example, two pictures of ilmenite and alien meteorite are shown in Fig. 1(c) and Fig. 1(d). Ilmenite is a known mineral on earth, while meteorite contains 50% of unknown chemical elements (Unk. in Fig. 1(c) and (d)). After detection in proportions of all known chemical elements, they share the same chemical formulas of $FeTiO_3$(20% Fe, 20% Ti and 60% O), leading to the same LD, which is obviously unreasonable. In this case, we have no choice but to introduce the background concentration for representing the proportion of unknown elements because they cannot be detected yet. To give another example, Fig. 2(a) and Fig 2(d) are two images of human face. Despite they have significant difference in the intensity of emotion, two identical LDs in Fig. 2(b) and Fig. 2(e) are assigned to them due to the similarity of the proportion of each emotions. In this case, each of the labels describes a specific emotion (happiness, sadness, etc.), so

the background concentration term represents the description degree of "no emotion", which is abstract and hard to be measured. In other words, the stronger the emotion, the lower the background concentration it should be assigned. Appending background concentration terms to the LDs in Fig. 2(b) and Fig. 2(e), they become concentration distributions in Fig. 2(c) and Fig. 2(f), which help us to distinguish the two images in Fig. 2(a) and Fig. 2(d) very well.

In this paper, we come up with a novel model, named CDL-LD. With probabilistic methods and neural networks, it can learn concentration distributions from the existing label distribution dataset. The main contributions of our model are summarized as follows:

- Based on real-world examples, we come up with a new learning paradigm named concentration distribution learning, which overcomes the ambiguity of LDs in traditional LDL methods. In addition, excavating the background concentration makes full use of the information in the datasets and benefits the downstream tasks.

- With the original version of an LDL dataset, we construct the first real-world concentration distribution dataset, which further proves the realistic significance of background concentration and the effectiveness of our method in concentration distribution learning.

- Although concentration distribution learning is an excellent paradigm, there is no existing concentration distribution dataset for learning. Our method learns CDs directly from existing LDL datasets, which improves the universality of CDL.

- We conduct extensive experiments to validate the advantages of our methods over the baseline algorithms on concentration distribution learning on LDL datasets. Furthermore, by using the Rademacher complexity, we show the generalization bound of the proposed model and theoretically prove that it is possible to directly construct a CDL model from the LDL datasets for the first time.

Extensive experiments on the benchmark datasets clearly show that the LD predicted by our method is better than that predicted by state-of-the-art LDL methods, and the recovery results of our method on background concentrations also fit well with reality.

## 2. Related Works

### 2.1. Label Distribution Learning

LDL (Kou et al., 2024; 2025; ?; Wang et al., 2025) is a new machine learning paradigm that constructs a model to predict the label distribution of samples. At first, LDL were achieved through problem transformation that transforms LDL into a set of single label learning problems such as PT-SVM, PT-Bayes (Geng, 2016), or through algorithm adaptation that adopts the existing machine learning algorithms to LDL, such as AA-kNN and AA-BP (Geng, 2016). SA-IIS (Geng, 2016) is the first model that specially designed for LDL, whose objective function is a mixture of maximum entropy loss (Berger et al., 1996) and KL-divergence. Based on SA-IIS, SA-BFGS (Geng, 2016) adopts BFGS to optimize the loss function, which is faster than SA-IIS. In addition, LDLLC (Zheng et al., 2018) leverages local label correlation to ensure that prediction distributions between similar instances are as close as possible. LCLR (Ren et al., 2019) first models the global label correlation using a low-rank matrix and then updates the matrix on clusters of samples to consider local label correlation. LDL forests (LDLFs) (Shen et al., 2017) is based on differentiable decision trees and may be combined with representation learning to provide an end-to-end learning framework. LDLLDM (Wang & Geng, 2023) can handle incomplete label distribution learning and learn the global and local label distribution manifolds to take advantage of label correlations.

### 2.2. Concentration Distribution Learning

Although the above LDL methods have achieved great success, none of them take the background concentration into account. Our work is the first to introduce the concept of background concentration to traditional label distribution learning and to develop the novel paradigm of concentration distribution learning. Further experiments on various real-world datasets also prove the effectiveness of our method on learning concentration distributions. If the proposed method can precisely predict concentration distributions from label distribution datasets, it can better excavate the hidden information in these datasets and benefit the downstream tasks.

## 3. Proposed Model

**Notations**: Let $n$, $m$ and $c$ represent the number of samples, the dimension of features, and the number of labels. Let $\boldsymbol{x} \in \mathbb{R}^m$ denote a feature vector and $\boldsymbol{y} \in \mathbb{R}^c$ denote its corresponding ground-truth label distribution vector, which satisfies $\forall i \in [1, 2, ..., c]$, $y_i \in [0, 1]$ and $\sum_{i=1}^{c} y_i = 1$. The feature matrix and the corresponding ground-truth label distribution matrix can be denoted by $\mathbf{X} = [\boldsymbol{x}_1; \boldsymbol{x}_2; \ldots; \boldsymbol{x}_n] \in \mathbb{R}^{n \times m}$ and $\mathbf{Y} = [\boldsymbol{y}_1; \boldsymbol{y}_2; \ldots; \boldsymbol{y}_n] \in \mathbb{R}^{n \times c}$, respectively. We aim to learn from the dataset $\mathcal{D} = [\mathbf{X}; \mathbf{Y}]$ and predict concentration distribution vectors $\boldsymbol{c}_d = [\boldsymbol{b}, \mu] \in \mathbb{R}^{c+1}$ for unseen instances. It is composed of the real label distribution vector $\boldsymbol{b}$ and the background concentration term $\mu$, in which $\boldsymbol{b} \in \mathbb{R}^c$, $\mu > 0$, $\forall i \in [1, 2, ..., c]$, $b_i \in [0, 1]$ and $\sum_{i=1}^{c} b_i + \mu = 1$.

First, to relate label distributions and concentration distributions, we define $\boldsymbol{p} \in \mathbb{R}^c$ as the apparent label distribution vector of the corresponding concentration distribution vector $\boldsymbol{c}_d = [\boldsymbol{b}, \mu] \in \mathbb{R}^{c+1}$, where $\forall i \in [1, 2, ..., c]$, $p_i \in [0, 1]$ and $\sum_{i=1}^{c} p_i = 1$. The relationship between the apparent label distribution vector and the concentration distribution vector can be formulated as:

$$\boldsymbol{p} = \boldsymbol{b} + \boldsymbol{\mu}^*, \tag{1}$$

where $\boldsymbol{\mu}^* \in \mathbb{R}^c$, $\forall i \in [1, 2, ..., c]$, $\boldsymbol{\mu}^*_i \in [0, 1]$ and $\sum_{i=1}^{c} \boldsymbol{\mu}^*_i = \mu$. Eq. (1) indicates that the distribution of the background concentration $\mu$ on the real label distribution vector $\boldsymbol{b}$ converts the concentration distributions to the apparent label distributions. In other words, the apparent label distribution is affected by both the dataset and the background concentration.

From a perspective of probability, we assume that $\boldsymbol{p}$ obeys the Dirichlet distribution, i.e., $\boldsymbol{p} \sim Dir(\boldsymbol{\alpha})$, where $\boldsymbol{\alpha} \in \mathbb{R}^c$ is the vector of distribution parameters. According to the expectation formula of Dirichlet distribution, the expectation on $p_i$ can be written as

$$\mathbb{E}_{p_i} = \frac{\alpha_i}{\sum_{j=1}^{c} \alpha_j}. \tag{2}$$

From the form of Eq. (2), the distribution parameter $\alpha_i$ can be regarded as the belief mass on the $i$-th class, i.e.,

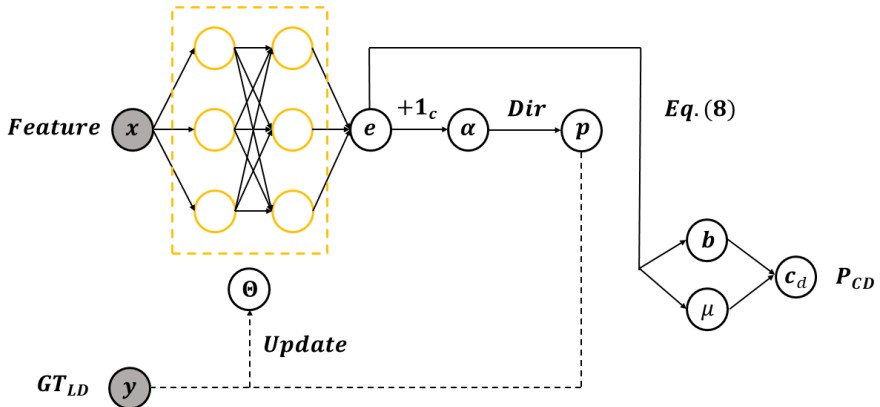

Figure 3. The framework of our method. The feature vector $\boldsymbol{x}$ and network parameter matrix $\Theta$ produce dataset-side confidence vector $\boldsymbol{e}$ by forward propagation of the nerual network, and $\Theta$ is updated by backward propagation with the ground-truth label distribution vector $\boldsymbol{y}$ and the apparent label distribution vector $\boldsymbol{p}$. $\boldsymbol{p}$ is sampled in the Dirichlet distribution of parameters vector $\boldsymbol{\alpha} = \boldsymbol{e} + \boldsymbol{1}_c$, where $\boldsymbol{1}_c$ is all-ones vector of size $c$. With Eq. (8), the real label distribution vector $\boldsymbol{b}$ and the background concentration $\mu$ are generated from the confidence vector $\boldsymbol{e}$. Finally we get the predicted concentration distribution vector $\boldsymbol{c}_d = [\boldsymbol{b}, \mu]$.

the degree of confidence that the corresponding instance should be classified into the $i$-th class. This confidence comes from two sources, one is the certain part extracted from the dataset, and the other is the hidden part decided by the background concentration, i.e.,

$$\alpha_i = e_i + u_i, \tag{3}$$

where $e_i > 0$ and $u_i > 0$ represents the degree of confidence on the $i$-th class from the dataset and the background concentration respectively. Denote $\boldsymbol{0}_c, \boldsymbol{1}_c \in \mathbb{R}^c$ as an all-zeros vector and an all-ones vector. Based on reality, when there is nothing in the dataset to provide confidence, i.e., $\forall i \in [1, 2, ..., c], \boldsymbol{e} = [e_1, e_2, ..., e_c] = \boldsymbol{0}_c$, $\boldsymbol{p}$ should be a uniform distribution. In this case, the vector of distribution parameters $\boldsymbol{\alpha} = \boldsymbol{1}_c$ and $Dir(\boldsymbol{\alpha})$ degrades to a flat Dirichlet distribution. Substitute $\boldsymbol{e} = \boldsymbol{0}_c$ and $\boldsymbol{\alpha} = \boldsymbol{1}_c$ in Eq. (3), we have

$$\begin{aligned} \boldsymbol{1}_c &= \boldsymbol{0}_c + \boldsymbol{u} \\ \boldsymbol{u} &= \boldsymbol{1}_c, \end{aligned} \tag{4}$$

where $\boldsymbol{u} = [u_1, u_2, ..., u_c]$. Substitute Eq. (4) in Eq. (3), the relation between $\alpha_i$ and $e_i$ is formulated as

$$\alpha_i = e_i + 1. \tag{5}$$

Finally, substitute $\alpha_i$ of Eq. (5) in Eq. (2),

$$\begin{aligned} \mathbb{E}_{p_i} &= \frac{e_i + 1}{\sum_{j=1}^{c}(e_j + 1)} \\ &= \frac{e_i + 1}{\sum_{j=1}^{c} e_j + c}. \end{aligned} \tag{6}$$

In Eq. (1), assuming that the background concentration is evenly spread on each class of the real label distribution vector $\boldsymbol{b}$ in probability, i.e., $\forall i \in [1, 2, ..., c], \mathbb{E}_{\mu_i^*} = \frac{\mu}{c}$, then another form of the expectation of $p_i$ can be expressed as

$$\mathbb{E}_{p_i} = b_i + \frac{\mu}{c}. \tag{7}$$

Combining Eq. (6) and Eq. (7), the expressions of $b_i$ and $\mu$ can be obtained as

$$\begin{aligned} \frac{e_i + 1}{\sum_j (e_i + 1)} &= b_i + \frac{\mu}{c} \\ b_i = \frac{e_i}{\sum_{j=1}^{c} e_j + c}, &\quad \mu = \frac{c}{\sum_{j=1}^{c} e_j + c} \end{aligned} \tag{8}$$

To obtain the vector of dataset-side confidence $\boldsymbol{e}$, we apply a confidence network $\boldsymbol{e} = f(\boldsymbol{x}|\Theta)$ on the dataset $\mathcal{D} = [\mathbf{X}; \mathbf{Y}]$, where $\boldsymbol{x} \in \mathbb{R}^m$ is an instance from $\mathbf{X}$, $f$ is the network function and $\Theta$ is its parameters. Specifically, the softmax layer of a conventional neural network is replaced with an activation function layer (i.e., RELU (Glorot et al., 2011)) to ensure that the network outputs non-negative values, which are considered as the confidence vector $\boldsymbol{e} = [e_1, e_2, ..e_c]$. Accordingly, the vector of Dirichlet distribution parameters $\boldsymbol{\alpha}$ can be obtained.

For each instance, in conventional neural network, the MSE (Mean Square Error) loss is usually employed as

$$\mathcal{L}_{MSE} = ||\boldsymbol{y} - \boldsymbol{p}||_2^2, \tag{9}$$

where $\boldsymbol{y}$ and $\boldsymbol{p}$ are the ground-truth and the predicted label distribution vector of the instance respectively. For our

model, given the degree of dataset-side confidence $e$ of the instance obtained through the confidence network, we can get the parameter $\boldsymbol{\alpha}$ (i.e., $\alpha_i = e_i + 1$) of the Dirichlet distribution and sample $\boldsymbol{p}$ from $Dir(\boldsymbol{\alpha})$. After a simple modification on Eq. (9), we have the adjusted MSE loss of our model:

$$
\begin{aligned}
\mathcal{L}_{AMSE}(\boldsymbol{\alpha}) &= \int \|\boldsymbol{y} - \boldsymbol{p}\|_2^2 \frac{1}{B(\boldsymbol{\alpha})} \prod_{i=1}^{c} p_i^{\alpha_i - 1} d\boldsymbol{p} \\
&= \sum_{i=1}^{c} \underbrace{(y_i - \frac{\alpha_i}{S})^2}_{\mathcal{L}_{err}} + \underbrace{\frac{\alpha_i(S - \alpha_i)}{S^2(S+1)}}_{\mathcal{L}_{var}} \\
&= \sum_{i=1}^{c} (y_i - \hat{p}_i)^2 + \frac{\hat{p}_i(1 - \hat{p}_i)}{S+1},
\end{aligned}
\tag{10}
$$

where $S = \sum_{i=1}^{c} \alpha_i$ and $\hat{p}_i = \frac{\alpha_i}{S}$. By decomposing the first and second moments, the loss aims to achieve the joint goals of minimizing the predictive error and the variance of the Dirichlet distribution generated by the neural network specifically for each sample in the training set. While doing so, it prioritizes data fit over variance estimation. With the trained neural network $e = f(\boldsymbol{x}|\Theta)$ and then substitute $e$ in Eq. 8, we get the predicted concentration distribution vector $\boldsymbol{c}_d = [\boldsymbol{b}, \mu]$. The overall probabilistic graph model of CDL-LD is presented in Fig. 3.

### 3.1. Generalization Bound

In this subsection, we first provide Rademacher complexity (Mohri et al., 2012) for CDL-LD, which is a commonly used tool for comprehensive analysis of data-dependent risk bounds.

**Definition 1.** *Let $\mathcal{H}$ be a family of functions mapping from $\mathcal{X}$ to [0,1] and $\mathcal{S}$ be a set of fixed samples with size $n$. Then, the empirical Rademacher complexity of $\mathcal{H}$ with respective to $\mathcal{S}$ is defined as*

$$
\begin{aligned}
\widehat{\mathcal{R}}_S(\mathcal{H}) &= \mathbb{E}_{\boldsymbol{\sigma}} \left[ \sup_{h \in \mathcal{H}} \frac{1}{n} \sum_{i=1}^{n} \sigma_i h(x_i) \right] \\
&\leq \mathbb{E}_{\boldsymbol{\sigma}} \left[ \frac{1}{n} \sum_{i=1}^{n} \sigma_i \mathcal{L}_{AMSE}(\alpha_i) \right],
\end{aligned}
\tag{11}
$$

*where $\sigma_i \in [0, 1]$ and $\alpha_i$ represents the Dirichlet distribution parameter vector of the $i$-th instance.*

**Lemma 1.** *Let $\mathcal{H}$ be a family of functions. For any $\delta > 0$, with probability at least $1 - \delta$, for all $h \in \mathcal{H}$ such that*

$$
\mathcal{L}(h) \leq \mathcal{L}_S(h) + \widehat{\mathcal{R}}_S(\mathcal{H}) + 3\sqrt{\frac{\log 2/\delta}{2n}},
\tag{12}
$$

*where $\mathcal{L}(h)$ and $\mathcal{L}_S(h)$ are the generalization risk and empirical risk with respective to h, and $n$ represents the number of instances.*

Substitute Eq. (11) in Eq. (12), we have

$$
\begin{aligned}
\mathcal{L}(h) \leq &\mathcal{L}_S(h) + \mathbb{E}_{\boldsymbol{\sigma}} \left[ \frac{1}{n} \sum_{i=1}^{n} \sigma_i \mathcal{L}_{AMSE}(\alpha_i) \right] \\
&+ 3\sqrt{\frac{\log 2/\delta}{2n}}.
\end{aligned}
\tag{13}
$$

According to the second line of Eq. (10) and $S = \sum_{i=1}^{c} \alpha_i$, there holds that $\mathcal{L}_{AMSE}(\boldsymbol{\alpha}) = \sum_{i=1}^{c}(y_i - \frac{\alpha_i}{S})^2 + \frac{\alpha_i(S - \alpha_i)}{S^2(S+1)} > 0$. With $\sigma_i \in [0, 1]$, the expectation term in Eq. (13) can be simplified as

$$
\begin{aligned}
\mathbb{E}_{\boldsymbol{\sigma}} \left[ \frac{1}{n} \sum_{i=1}^{n} \sigma_i \mathcal{L}_{AMSE}(\alpha_i) \right] &\leq \frac{1}{n} \sum_{i=1}^{n} \mathcal{L}_{AMSE}(\alpha_i) \\
&\leq \frac{1}{n} \sum_{i=1}^{n} \sum_{j=1}^{c} \left[ (y_{ij} - \frac{\alpha_{ij}}{S_i})^2 + \frac{\alpha_{ij}(S_i - \alpha_{ij})}{S_i^2(S_i + 1)} \right] \\
&\leq \frac{1}{n} \sum_{i=1}^{n} \sum_{j=1}^{c} \left[ (y_{ij} - \frac{\alpha_{ij}}{S_i})^2 + \frac{\frac{\alpha_{ij}}{S_i}(1 - \frac{\alpha_{ij}}{S_i})}{S_i(S_i + 1)} \right] \\
&\leq \frac{1}{n} \sum_{i=1}^{n} \left[ 1 + \frac{1}{4c(c+1)} \right]
\end{aligned}
\tag{14}
$$

Substitute Eq. (14) in Eq. (13), we get

$$
\mathcal{L}(h) \leq \mathcal{L}_S(h) + \frac{1}{n} \sum_{i=1}^{n} \left[ 1 + \frac{1}{4c(c+1)} \right] + 3\sqrt{\frac{\log 2/\delta}{2n}}.
\tag{15}
$$

Let $n$ in Eq. (15) tends to infinity:

$$
\mathcal{L}(h) \leq \mathcal{L}_S(h) + \underbrace{\left[ 1 + \frac{1}{4c(c+1)} \right]}_{bound}.
\tag{16}
$$

From Eq. (16), we can observe that when the number of instances $n$ tends to infinity, the generalization risk of our proposed model will be upper-bounded by the empirical risk of it with a bound of $\frac{1}{4c(c+1)}$, which indicates that this bound shrinks when the number of classes increases. **This conclusion is intuitive because the background concentration tends to zero when $c$ tends to infinity, degrading the CDL problem to learnable LDL problem.** Combining all equations together, we can draw the conclusion that it is possible to directly construct a CDL model from the LDL datasets, and the concentration distribution learning problem itself is also learnable.

# 4. Experiments

## 4.1. Experimental Configurations

### 4.1.1. EXPERIMENTAL DATASETS

The experiments are carried out on 12 real-world datasets with label distribution. The statistics of these datasets are summarized in Table 1. Among these, the first eight (from Alpha to Spoem) are from the clustering analysis of genome-wide expression in Yeast Saccharomyces cerevisiae (Eisen et al., 1998). The SJAFFE is collected from JAFFE (Lyons et al., 1998), and the SBU_3DFE is obtained from BU_3DFE (Yin et al., 2006). The Scene consists of multi-label images, where the label distributions are transformed from rankings (Geng & Xia, 2014). SJA_c (in section 4.2) is the first CDL dataset generated from SJAFFE dataset in this paper. For each dataset, the last class of the ground-truth label distribution is regarded as the background concentration and will be invisible in training.

*Table 1.* Statistics of the 12 datasets

| Dataset (abbr.) | # Instances | # Features | # Labels |
|---|---|---|---|
| Alpha (Alp) | 2465 | 24 | 18 |
| Cdc (Cdc) | 2465 | 24 | 15 |
| Cold (Col) | 2465 | 24 | 4 |
| Diau (Dia) | 2465 | 24 | 7 |
| Elu (Elu) | 2465 | 24 | 14 |
| Heat (Hea) | 2465 | 24 | 6 |
| Spo (Spo5) | 2465 | 24 | 6 |
| Spo5 (Spo) | 2465 | 24 | 3 |
| SJAFFE (SJA) | 213 | 243 | 6 |
| SBU_3DFE (SBU) | 2500 | 243 | 6 |
| Scene (Sce) | 2000 | 294 | 9 |
| SJA_c (SJAc) | 213 | 243 | 7 |

*Table 2.* Formulas of the four evaluation metrics. Here ↓ indicates that smaller values are better, and ↑ indicates that larger values are better.

| Measure | Formula |
|---|---|
| Chebyshev↓ | $Dis_1(D, \hat{D}) = \max_j \lvert d_j - \hat{d}_j \rvert$ |
| Clark↓ | $Dis_2(D, \hat{D}) = \sqrt{\sum_{j=1}^{c} \frac{(d_j - \hat{d}_j)^2}{(d_j + \hat{d}_j)^2}}$ |
| KL↓ | $Dis_3(D, \hat{D}) = \sum_{j=1}^{c} d_i \ln \frac{d_i}{\hat{d}_i}$ |
| Cosine↑ | $Sim_1(D, \hat{D}) = \frac{\sum_{j=1}^{c} d_j \hat{d}_j}{\sqrt{\sum_{j=1}^{c} d_j^2}\sqrt{\sum_{j=1}^{c} \hat{d}_j^2}}$ |

### 4.1.2. EVALUATION METRICS

As suggested in (Geng, 2016), we adopt three distance metrics (i.e., Chebyshev, Clark and KL) and one similarity metric (i.e. Cosine) to evaluate the performances of methods. The formulas of these evaluation metrics are summarized in Table 2.

### 4.1.3. COMPARISON METHODS

We compare the proposed methods with four baseline LDL methods, including LDLLC, SA-IIS, LCLR, LDLFs, LDLLDM and DLDL, which are briefly introduced as below.

- LDLLC (Zheng et al., 2018): LDLLC makes use of local label correlation to guarantee that prediction distributions between comparable instances are as close as possible.

- SA-IIS (Geng, 2016): SA-IIS learns the label distribution using KL divergence and the maximum entropy model.

- LCLR (Ren et al., 2019): LCLR uses a low-rank matrix to model global label correlation first, and then it updates the matrix on sample clusters to account for local label correlation.

- LDLFs (Shen et al., 2017): To provide an end-to-end learning framework, LDL forests (LDLFs), which are based on differentiable decision trees, can be coupled with representation learning.

- LDLLDM (Wang & Geng, 2023): LDLLDM learns both the global and local label distribution manifolds in order to take advantage of label correlations. It is also capable of handling partial label distribution learning.

- DLDL (Jia et al., 2024): DLDL unifies label enhancement (LE) and LDL into a joint model and avoids the drawbacks of the previous LE methods. Furthermore, it theoretically proves that directly learning an LDL model from logical labels is feasible.

The parameters of the methods are as follows. The suggested parameters are used for LDLLC, IIS-LLD, LCLR, LDLFs and DLDL. For LDLLDM, $\lambda_1, \lambda_2$ and $\lambda_3$ are tuned from $\{10^{-3}, 10^{-2}, ..., 10^3\}$, and $g$ is tuned from 1 to 14. Note that all the baseline methods are LDL algorithms, they are not able to provide concentration distributions. So, for each instance, we append $g + \delta$ as the predicted background concentration to the predicted label distribution vector, where $g$ is the ground-truth description degree of the last class and $-0.2g < \delta < 0.2g$ is a random noise to simulate the inaccurate learning of background concentration in the baseline methods. Then normalization will be applied on

this concentration distribution vector to ensure that all of its elements sums up to 1. We run each method for ten-fold cross-validation.

## 4.2. Construction of CDL dataset

The generation mechanism of the SJAFFE dataset makes it a potential concentration distribution dataset. Specifically, the six classes (happy, sad, surprised, angry, disappointed, and fear) of the original SJAFFE dataset are semantic ratings averaged over 60 Japanese female subjects. A 5-level scale was used for each of the six adjectives (5 for highest and 1 for lowest). An example of the rating vector in the original SJAFFE dataset can be $s = [3, 4.8, 1.2, 2.1, 2.4, 1.5]$, and after normalization, it becomes a label distribution vector $s_n$

$$s_n = \frac{[3, 4.8, 1.2, 2.1, 2.4, 1.5]}{3 + 4.8 + 1.2 + 2.1 + 2.4 + 1.5} \qquad (17)$$
$$= [0.2, 0.32, 0.08, 0.14, 0.16, 0.1].$$

According to the definition of background concentration, we have $\mu = 5 * 6 - \sum_{i=1}^{6} s_i$, where $\mu$ is regarded the ground-truth background concentration and $s_i$ is the $i$-th element of the original SJAFFE rating vector. In the case above, $\mu = 5 * 6 - (3 + 4.8 + 1.2 + 2.1 + 2.4 + 1.5) = 15$. Appending $\mu$ to the last of the original rating vector $s$ and then applying normalization on the new vector, we get a standard concentration distribution vector $c_d$

$$c_d = \frac{[3, 4.8, 1.2, 2.1, 2.4, 1.5, 15]}{3 + 4.8 + 1.2 + 2.1 + 2.4 + 1.5 + 15} \qquad (18)$$
$$= [0.1, 0.16, 0.04, 0.07, 0.08, 0.05, 0.5].$$

Repetition of the steps above on each instance in the SJAFFE dataset gives the ground-truth concentration distributions of them, **making it the first concentration distribution dataset named SJA_c.**

## 4.3. Results and Discussion

Table 3 presents the experimental results (mean±std) of all the 7 algorithms on 12 datasets in terms of Chebyshev (abbr. Cheby), Clark, KL, and Cosine, with the best results highlighted in boldface. According to the results recorded in Table 3, we observe that

- Our method CDL-LD achieves the lowest average rank in terms of all of the four evaluation metrics. Specifically, out of the 48 statistical comparisons, CDL-LD ranks 1st in 93.75% (45 out of 48) cases. In general, CDL-LD performs better than most comparison algorithms.

- On some LDL datasets, our method achieves superior performance. For example, on Col, Dia, Spo and Scene, CDL-LD significantly improves the results compared

with the other baseline algorithms in terms of all the four metrics.

- On the concentration distribution dataset (i.e, SJA_c), CDL-LD shows overwhelming superiority to the other baseline algorithms in all of the four metrics. This result proves the effectiveness of CDL-LD in real-world concentration distribution learning problems.

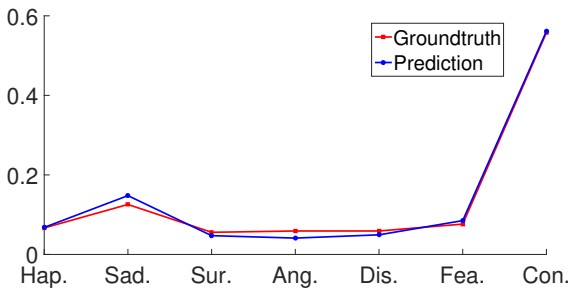

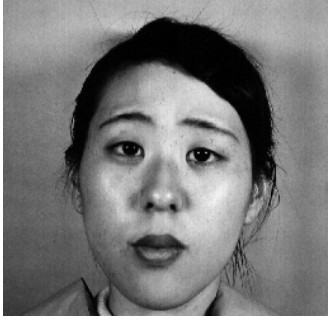

*Figure 4.* The visualization of a typical result of our method on the SJA_c dataset and its corresponding image.

Additionally, the visualization of a typical result on the SJA_c dataset is presented in Fig. 4, where the scales on the horizontal axis represent six emotions of happiness, sadness, surprise, anger, disappointment, and fear, while the last represents the background concentration. In this figure, we can observe that the predictive result of our method is very close to the ground-truth concentration distribution in the first six classes (the real label distributions of six emotions) and in the last class (the background concentration of "no emotion"). Specifically, both the ground-truth and the predicted CDs indicate the emotion of faint sadness (a peak on sadness with high background concentration), matching the corresponding image very well. This further proves that background concentrations exist in reality and our method has the ability to precisely excavate background concentrations from real-world datasets.

To sum up, the experimental results further support the competitive performance of the proposed algorithm, indicating its effectiveness in real-world concentration distribution learning problems.

| | Metric | CDL-LD | LDLLC | SA-IIS | LCLR | LDLFs | LDLLDM | DLDL |
|---|---|---|---|---|---|---|---|---|
| Alp | Cheby | **0.0245±.0002(1)** | 0.0571±.0022(7) | 0.0366±.0028(2) | 0.0529±.0018(6) | 0.0502±.0005(3) | 0.0523±.0013(5) | 0.0520±.0002(4) |
| | Clark | **0.2797±.0009(1)** | 0.4586±.0128(6) | 0.7340±.0297(7) | 0.4410±.0002(5) | 0.4253±.0066(2) | 0.4365±.0211(4) | 0.4318±.0049(3) |
| | KL | **0.0098±.0000(1)** | 0.0914±.0018(7) | 0.0623±.0487(4) | 0.0858±.0020(6) | 0.0835±.0005(5) | 0.0365±.0028(3) | 0.0351±.0001(2) |
| | Cosine | **0.9894±.0017(1)** | 0.9560±.0019(7) | 0.9824±.0027(2) | 0.9563±.0021(6) | 0.9590±.0007(3) | 0.9570±.0009(5) | 0.9573±.0001(4) |
| Cdc | Cheby | **0.0291±.0002(1)** | 0.0619±.0068(4) | 0.1456±.0119(7) | 0.0565±.0022(2) | 0.0632±.0003(6) | 0.0599±.0008(3) | 0.0626±.0037(5) |
| | Clark | **0.2891±.0003(1)** | 0.4876±.0270(6) | 0.8329±.0030(7) | 0.4327±.0007(2) | 0.4539±.0026(4) | 0.4457±.0122(3) | 0.4611±.0086(5) |
| | KL | **0.0122±.0003(1)** | 0.1114±.0082(6) | 0.1327±.0506(7) | 0.0959±.0010(4) | 0.1031±.0007(5) | 0.0426±.0041(2) | 0.0448±.0041(3) |
| | Cosine | **0.9870±.0005(1)** | 0.9485±.0079(5) | 0.8489±.0164(7) | 0.9565±.0024(2) | 0.9479±.0003(6) | 0.9530±.0007(3) | 0.9502±.0047(4) |
| Col | Cheby | **0.0821±.0009(1)** | 0.1606±.0077(7) | 0.1561±.0312(6) | 0.1475±.0084(4) | 0.1488±.0030(5) | 0.1441±.0013(2) | 0.1454±.0060(3) |
| | Clark | **0.2073±.0002(1)** | 0.4062±.0181(6) | 0.4670±.1468(7) | 0.3734±.0218(4) | 0.3849±.0015(5) | 0.3674±.0064(2) | 0.3719±.0146(3) |
| | KL | **0.0262±.0006(1)** | 0.2421±.0135(7) | 0.1051±.0599(4) | 0.2180±.0151(5) | 0.2209±.0044(6) | 0.0973±.0064(2) | 0.0985±.0100(3) |
| | Cosine | **0.9737±.0006(1)** | 0.9139±.0058(7) | 0.9385±.0219(2) | 0.9266±.0056(6) | 0.9277±.0022(5) | 0.9302±.0015(3) | 0.9289±.0044(4) |
| Dia | Cheby | **0.0677±.0004(1)** | 0.1092±.0014(4) | 0.1529±.0755(7) | 0.1140±.0004(6) | 0.1105±.0100(5) | 0.1079±.0033(3) | 0.1072±.0111(2) |
| | Clark | **0.2910±.0001(1)** | 0.4420±.0055(6) | 0.5501±.0090(7) | 0.4394±.0006(5) | 0.4308±.0219(4) | 0.4186±.0058(2) | 0.4227±.0188(3) |
| | KL | **0.0276±.0003(1)** | 0.1707±.0046(6) | 0.0961±.0236(4) | 0.1721±.0010(7) | 0.1689±.0114(5) | 0.0756±.0010(3) | 0.0732±.0098(2) |
| | Cosine | **0.9718±.0003(1)** | 0.9292±.0001(5) | 0.9042±.0677(7) | 0.9259±.0004(6) | 0.9294±.0102(4) | 0.9315±.0028(2) | 0.9344±.0105(3) |
| Elu | Cheby | **0.0317±.0001(1)** | 0.0657±.0014(6) | 0.0506±.0150(2) | 0.0647±.0003(5) | 0.0629±.0012(3) | 0.0632±.0008(4) | 0.0683±.0003(7) |
| | Clark | **0.2740±.0002(1)** | 0.4592±.0105(6) | 0.6535±.2047(7) | 0.4333±.0116(4) | 0.4273±.0013(2) | 0.4316±.0022(3) | 0.4505±.0005(5) |
| | KL | **0.0119±.0001(1)** | 0.1113±.0028(7) | 0.0587±.0409(4) | 0.1027±.0026(6) | 0.1001±.0006(5) | 0.0433±.0009(2) | 0.0487±.0001(3) |
| | Cosine | **0.9871±.0001(1)** | 0.9479±.0025(6) | 0.9742±.0114(2) | 0.9507±.0004(5) | 0.9529±.0008(3) | 0.9523±.0012(4) | 0.9471±.0007(7) |
| Hea | Cheby | **0.0641±.0004(1)** | 0.1183±.0037(5) | 0.1006±.0884(2) | 0.1205±.0004(6) | 0.1154±.0081(4) | 0.1276±.0028(7) | 0.1138±.0042(3) |
| | Clark | **0.2399±.0002(1)** | 0.4132±.0131(6) | 0.3095±.1641(2) | 0.4112±.0009(5) | 0.3979±.0219(3) | 0.4290±.0061(7) | 0.3997±.0106(4) |
| | KL | **0.0218±.0001(1)** | 0.1828±.0102(7) | 0.0419±.0429(2) | 0.1800±.0019(6) | 0.1730±.0109(5) | 0.0925±.0031(4) | 0.0780±.0037(3) |
| | Cosine | **0.9776±.0002(1)** | 0.9309±.0034(5) | 0.9505±.0547(2) | 0.9307±.0005(6) | 0.9359±.0068(4) | 0.9247±.0024(7) | 0.9370±.0041(3) |
| Spo | Cheby | **0.0634±.0016(1)** | 0.1324±.0047(6) | 0.2541±.0232(7) | 0.1210±.0005(2) | 0.1251±.0036(3) | 0.1270±.0012(5) | 0.1254±.0039(4) |
| | Clark | **0.2584±.0035(1)** | 0.4611±.0079(6) | 0.6960±.0367(7) | 0.4228±.0017(2) | 0.4364±.0110(4) | 0.4355±.0008(3) | 0.4439±.0132(5) |
| | KL | **0.0254±.0007(1)** | 0.2140±.0065(7) | 0.2130±.0222(6) | 0.1914±.0028(4) | 0.2004±.0084(5) | 0.0927±.0024(2) | 0.0981±.0009(3) |
| | Cosine | **0.9754±.0008(1)** | 0.9190±.0044(6) | 0.8423±.0168(7) | 0.9301±.0010(2) | 0.9265±.0028(3) | 0.9253±.0028(4) | 0.9241±.0017(5) |
| Spo5 | Cheby | **0.3101±.0017(1)** | 0.3936±.0072(7) | 0.3850±.0275(6) | 0.3207±.0072(3) | 0.3202±.0073(2) | 0.3287±.0065(4) | 0.3489±.0093(5) |
| | Clark | **0.7726±.0044(1)** | 0.9373±.0049(7) | 0.8799±.0624(6) | 0.8372±.0123(3) | 0.8393±.0116(4) | 0.8548±.0034(5) | 0.7924±.0252(2) |
| | KL | **0.4343±.0151(1)** | 1.6749±.0256(7) | 0.7430±.0877(3) | 0.8877±.0236(6) | 0.8850±.0195(5) | 0.6660±.0054(2) | 0.7659±.0352(4) |
| | Cosine | **0.8133±.0011(1)** | 0.7542±.0058(7) | 0.8026±.1758(5) | 0.8112±.0058(2) | 0.8095±.0065(4) | 0.8063±.0065(3) | 0.7844±.0088(6) |
| SJA | Cheby | 0.4335±.0002(7) | 0.3484±.0473(3) | 0.3686±.0326(4) | **0.3045±.0056(1)** | 0.3734±.0119(5) | 0.3388±.0219(2) | 0.3790±.0216(6) |
| | Clark | **0.9992±.0002(1)** | 1.1948±.0591(7) | 1.0976±.0239(4) | 1.0627±.0053(3) | 1.0373±.0722(2) | 1.1630±.0126(6) | 1.1508±.0736(5) |
| | KL | **0.4841±.0007(1)** | 1.5855±.0465(7) | 0.7308±.0122(2) | 0.7653±.0718(3) | 0.7871±.1415(4) | 1.1815±.0159(6) | 0.9323±.0807(5) |
| | Cosine | **0.8161±.0013(1)** | 0.7211±.0425(6) | 0.7038±.0316(7) | 0.7478±.0021(4) | 0.7587±.0110(3) | 0.7222±.0149(5) | 0.7597±.0398(2) |
| SBU | Cheby | 0.3362±.0008(3) | 0.3688±.0047(5) | 0.3511±.0213(4) | **0.2972±.0019(1)** | 0.3123±.0019(2) | 0.3723±.0151(6) | 0.3823±.0171(7) |
| | Clark | **0.9682±.0035(1)** | 1.1914±.0079(7) | 1.1503±.0263(5) | 0.9798±.0207(3) | 0.9757±.0024(2) | 1.1761±.0418(6) | 1.1245±.0010(4) |
| | KL | **0.4641±.0028(1)** | 1.4267±.0065(7) | 0.7209±.0103(2) | 0.7894±.0360(3) | 0.8296±.0114(4) | 1.2469±.0190(5) | 1.2848±.0215(6) |
| | Cosine | 0.6310±.0009(7) | 0.7060±.0044(4) | 0.7183±.0208(3) | **0.7578±.0008(1)** | 0.7499±.0013(2) | 0.7040±.0097(5) | 0.6922±.0255(6) |
| Sce | Cheby | **0.3948±.0011(1)** | 0.5289±.0033(4) | 0.5438±.0447(6) | 0.4789±.0021(3) | 0.4765±.0083(2) | 0.5363±.0154(5) | 0.5440±.0153(7) |
| | Clark | **1.9714±.0041(1)** | 2.4907±.0019(2) | 2.7042±.0292(7) | 2.5404±.0004(6) | 2.5387±.0037(5) | 2.5202±.0065(3) | 2.5237±.0333(4) |
| | KL | **0.4918±.0010(1)** | 1.1354±.0108(2) | 1.7242±.0340(7) | 1.5624±.0077(5) | 1.5938±.0605(6) | 1.2310±.1056(3) | 1.2955±.0116(4) |
| | Cosine | **0.6063±.0020(1)** | 0.5787±.0048(2) | 0.4772±.0116(6) | 0.5282±.0011(3) | 0.4629±.0083(7) | 0.5187±.0072(5) | 0.5198±.0234(4) |
| SJAc | Cheby | **0.1153±.0031(1)** | 0.3190±.0096(4) | 0.3024±.0014(3) | 0.2504±.0003(2) | 0.3598±.0069(7) | 0.3303±.0148(5) | 0.3518±.0065(6) |
| | Clark | **0.5336±.0048(1)** | 0.9786±.0076(5) | 0.8086±.0100(3) | 0.7821±.0076(2) | 1.1820±.0341(7) | 0.9712±.0124(4) | 0.9983±.0164(6) |
| | KL | **0.0722±.0020(1)** | 0.7183±.0020(6) | 0.2021±.0088(2) | 0.4076±.0017(3) | 1.0728±.0394(7) | 0.4573±.0346(4) | 0.5089±.0277(5) |
| | Cosine | **0.9740±.0062(1)** | 0.7459±.0194(4) | 0.8824±.0192(2) | 0.8434±.0046(3) | 0.7160±.0489(6) | 0.7195±.0154(5) | 0.6791±.0072(7) |

*Table 3.* Predictive results (mean±std) and ranks of our and baseline methods in terms of four metrics on 12 datasets, where the best results are highlighted in boldface.

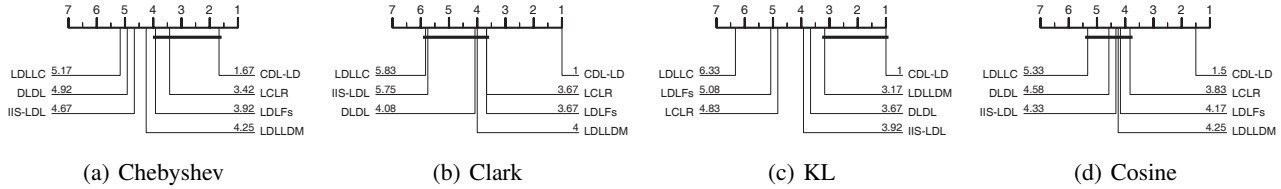

| | (a) Chebyshev | (b) Clark | (c) KL | (d) Cosine |

*Figure 5.* The CD diagram of CDL-LD against other six methods with the Bonferroni-Dunn test (CD = 2.3265 at 0.05 significance level).

| | CDL-LD | LDLLC | SA-IIS | LCLR | LDLFs | LDLLDM | DLDL |
|---|---|---|---|---|---|---|---|
| Cheby | **1.67(1)** | 5.17(7) | 4.67(5) | 3.42(2) | 3.92(3) | 4.25(4) | 4.92(6) |
| Clark | **1.00(1)** | 5.83(7) | 5.75(6) | 3.67(2) | 3.67(2) | 4.00(4) | 4.08(5) |
| KL | **1.00(1)** | 6.33(7) | 3.92(4) | 4.83(5) | 5.08(6) | 3.17(2) | 3.67(3) |
| Cosine | **1.50(1)** | 5.33(7) | 4.33(5) | 3.83(2) | 4.17(3) | 4.25(4) | 4.58(6) |

*Table 4.* Average ranks and their ranks of our and baseline methods in terms of four metrics on all the datasets, where the best average ranks are highlighted in boldface.

| Critical Value | Evaluation metric | Chebyshev | Clark | KL | Cosine |
|---|---|---|---|---|---|
| 2.913 | Friedman Statistics $F_F$ | 22.2645 | 40.2058 | 43.9416 | 21.8687 |

*Table 5.* Summary of the Friedman statistics $F_F$ in terms of four evaluation metrics, with the critical value at a significance level of 0.05 (7 algorithms on 12 datasets)

| | CDL-LD | LDLLC | SA-IIS | LCLR | LDLFs | LDLLDM | DLDL |
|---|---|---|---|---|---|---|---|
| Alp | **10.45(1)** | 177.12(6) | 69.34(2) | 78.75(3) | 136.89(5) | 114.36(4) | 180.71(7) |
| SBU | **29.96(1)** | 189.84(5) | 103.29(2) | 123.57(3) | 158.09(4) | 197.38(6) | 241.15(7) |
| Sce | **17.23(1)** | 186.58(6) | 92.62(3) | 92.03(2) | 143.62(4) | 172.51(5) | 234.02(7) |

*Table 6.* Average ranks and their ranks of our and baseline methods in terms of running time (unit: second) on three large datasets, where the best average ranks are highlighted in boldface.

### 4.4. Significance Tests

In this subsection, the average ranks of all methods are presented in Table 4. First, we conduct the Friedman test (Demšar, 2006) to study the comparative performance of all methods. Table 5 shows the Friedman statistics for each metric and the critical value. At the confidence level of 0.05, the null hypothesis that *all algorithms have equal performance* is rejected. Then we apply a post-hoc test, that is, the Bonferroni-Dunn test (Demšar, 2006) at the 0.05 significance level to test whether CDL-LD achieves significantly better performance compared to other algorithms. We use CDL-LD as the control algorithm with a critical difference (CD) (Demšar, 2006) to correct for the difference in mean level with the comparison algorithms. An algorithm is deemed to achieve significantly different performance from

CDL-LD if its average rank differs from that of CDL-LD by at least one critical difference.

The results are shown in Fig. 5. If the average rank of a comparing algorithm is within one CD to that of CDL-LD, they are connected with a thick line; otherwise, it is considered to have a significantly different performance from CDL-LD. It is impressive that CDL-LD achieves the lowest rank in terms of all four evaluation metrics, and the effectiveness of it is also more significant than all the other baseline methods based on Clark and Cosine. and the effectiveness of it is also more significant than all the other baseline methods based on Clark and Cosine.

### 4.5. Time Complexity Analysis

In this section, three large datasets are selected from all the training datasets. We compare the running time of all the algorithms, and the results are shown in Table 6. The hardware configuration of the test machine is as follows: AMD EPYC 7K62 48-Cores CPU, 377G running memory and NVIDIA GeForce RTX 3090 GPU. It can be observed from the results that the proposed model CDL-LD is much lower in terms of time cost than all other baseline algorithms, further proving the superiority of it.

## 5. Conclusion

This paper presents a novel paradigm called concentration distribution learning (CDL). Specifically, concentration distributions are constructed by introducing background concentration terms in label distributions. Due to the lack of concentration distribution learning datasets, we propose an algorithm to learn concentration distribution from traditional label distribution learning (LDL) datasets. Extensive experiments validate the advantage of CDL-LD against other baseline algorithms in concentration distribution learning, confirming the effectiveness of our method in training a CDL model from LDL datasets. Moreover, we build the first CDL dataset, that is, SJA_c from the original SJAFFE dataset, and further prove the ability of our method to solve real-world CDL problems. Future work will continue to explore this innovative direction, focusing on concentration distribution learning.

## Impact Statement

This paper presents work which comes up with a novel paradigm of concentration distribution learning and aims to advance the field of label distribution learning in Machine Learning. There are many potential societal consequences of our work, none of which we feel must be specifically highlighted here.

## Funding Information

This work was supported by the National Natural Science Foundation of China under Grant U24A20322 and the Big Data Computing Center of Southeast University.

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
