# OpenReview forum: "Concentration Distribution Learning from Label Distributions"
_ICML.cc/2025/Conference — ICML 2025 poster_

### Official Review · Reviewer_uvxc · 2025-02-25

**Overall Recommendation:** 4

**Summary:**

This paper proposes Concentration Distribution Learning (CDL), a new variant of Label Distribution Learning (LDL). In CDL, in addition to the common label distribution that represents the descriptions of each class, there is a dimension that represents the background information. Based on an assumption about the generation process of the concentration distribution, an estimation method is proposed to determine the concentration distribution. Experimental results validate the effectiveness of the proposed method, and a new CDL dataset is also proposed.

**Claims And Evidence:**

The claim that traditional label distribution learning does not take the background into account is novel in the literature and also makes sense. It is reasonable that two images with different backgrounds may have similar label distributions, and traditional label distributions may not describe the images well. Therefore, I think the novelty of the proposed setting is good

**Essential References Not Discussed:**

There are no essential references that need to be discussed.

**Experimental Designs Or Analyses:**

- The experiments consist mainly of two parts. The first part is a synthetic experiment, where traditional LDL experimental datasets are used and the last dimension is considered as a background. Such an experimental setting may differ from real-world scenarios because the last dimension may not serve as the background and its generation assumption may not be consistent with the assumption proposed in this paper (Eq. (1)).

- For the second part, a new CDL dataset is constructed, and it is surprising to see that the results of the proposed method are very similar to the ground-truth label (Fig. (4)). I am curious with these good experimental results. Can the authors elaborate more on the success of the proposed method on this dataset? Also, it might be helpful to include more experimental results on more data points.

**Methods And Evaluation Criteria:**

- The proposed method can learn a good label distribution under the proposed assumption. However, I think that the proposed assumption may be a bit strong. First, it assumes that the background vector is an all-one vector, which may be a bit easy since the contribution of the background to each class may be equal.

- Second, it assumes that the concentration distribution is an addition of the previous label distribution and the new concentration distribution. This may be more complex in real applications. I think this point needs to be clarified.

**Other Comments Or Suggestions:**

- The writing of the paper can be improved and revised to be clearer.

- More justification of the data generation model and its practicality in real-world applications should be given.

- Theoretical analysis of the proposed method can be done to improve the paper.

I am willing to increase my score if my concerns can be addressed properly.

**Other Strengths And Weaknesses:**

### Strengths
- The novelty of the new CDL setting is good and practical for LDL literature. This may open new research directions for LDL.
- The proposed method is simple and effective, validated by extensive experimental results.

### Weaknesses
- The writing of the paper needs to be improved. There are some typos and unclear expressions in the paper. For example, the word "are" is missing in line 18 of the abstract. There are some similar problems in the main text, which should be carefully reviewed and revised.

- Some equations seem to be redundant. For example, the left column on page 4 contains some equations that can be greatly simplified, since only simple calculations are performed.

- The assumption of the background vector generation process is too simple, which may limit the applications in the real-world scenarios.

**Questions For Authors:**

- I have a question about the CDL examples in the introduction. Fig. 1 is quite intuitive that different images with significant differences in the background should have different label distributions. Fig. 2 is not similar because the difference, i.e. the expression, may not correspond to the background. It is the main object of the image. Therefore, I am not sure that the core idea of the paper that the background of the image is important can be applied.

**Relation To Broader Scientific Literature:**

Not available.

**Theoretical Claims:**

No theoretical claims are made in this paper.

---

> ### Author Rebuttal · Authors · 2025-04-01
>
> Thank you for your time and effort on reviewing our paper. In what follows, we will address your **questions** in detail.
>
> **Experimental Designs Or Analyses 1: The traditional LDL experimental datasets may differ from real-world scenarios because the last dimension may not serve as the background and its generation assumption may not be consistent with the assumption proposed in this paper (Eq. (1)).**
>
> The paradigm of concentration distribution learning is a new concept, so there are no available CDL datasets. To this end, we construct the first real-world CDL dataset in our paper and conduct experiments on it to verify the effectiveness of our model of exvacating background concentration. And by hiding the last dimension of LDL datasets, we transform LDL datasets into CDL datasets. If our model can recover this hidden dimension, which is regarded as the background concentration, its ability to learn concentration distribution can be proved.
>
> **Experimental Designs Or Analyses 2: Can the authors elaborate more on the success of the proposed method on the CDL dataset? Also, it might be helpful to include more experimental results on more data points.**
>
> Thanks for your advice. In the final version of this paper, we will present more results on the constructed CDL dataset and some other data points.
>
> **Weakness 1 \& 2: The writing of the paper needs to be improved. There are some typos and unclear expressions in the paper, and some equations seem to be redundant.**
>
> Thanks for pointing out our shortcomings. In the final version of our paper, we will carefully correct these errors and simplify our equations.
>
> **Weakness 3: The assumption of the background vector generation process is too simple, which may limit the applications in the real-world scenarios.**
>
> The paradigm of concentration distribution learning is a new concept raised by this paper. Since we have no further prior knowledge about the background concentration distribution pattern, the assumption that it is evenly spread across each class of the real label distribution is general. Furthermore, the experimental results also validate the rationality of this assumption.
>
> **Other Comments Or Suggestions 1: The writing of the paper can be improved and revised to be clearer.**
>
> Thanks for your advise. This paper will be carefully revised in its final version.
>
> **Other Comments Or Suggestions 2: More justification of the data generation model and its practicality in real-world applications should be given.**
>
> For the data generation model, please refer to **Weakness 3** of this response. Additionally, in the final version of this paper, we will present more real-world applications of concentration distribution learning.
>
> **Other Comments Or Suggestions 3: Theoretical analysis of the proposed method can be done to improve the paper.**
>
> Please refer to the response of **Weakness 1** to reviewer 3FJ2.
>
> **Question: Fig. 2 is not similar because the difference, i.e. the expression, may not correspond to the background. It is the main object of the image. Therefore, I am not sure that the core idea of the paper that the background of the image is important can be applied.**
>
> The "background concentration" is named after the background, but it is an abstract concept and does not actually refer to the background of the image. In the case of facial expression, the background concentration doesn't refer to any expression, and it represents the description degree of “no emotion”. In other words, the stronger the emotion is, the lower the background concentration it should be assigned.

---

> > ### Comment · Reviewer_uvxc · 2025-04-04
> >
> > Thanks for the rebuttal, which has addressed my concerns. Therefore, I have increased my score.

---

### Official Review · Reviewer_5CcH · 2025-02-28

**Overall Recommendation:** 3

**Summary:**

The paper proposes a new label distribution learning method based on a novel hypothesis. Hypothesis: In existing label distribution learning, there is an issue where the current labels within the label set cannot adequately describe the samples. Based on this hypothesis, the authors suggest that concentration can be used to describe the extent to which existing labels describe the samples, and accordingly, the label distribution is augmented and expanded. The authors, based on this hypothesis, expand and augment the label distribution and then design a label distribution learning algorithm tailored to this hypothesis, setting concentration as a part of all label distributions. Experiments show that the algorithm proposed in the paper performs well on almost all datasets used, possessing the optimal average ranking, and the gap in the average ranking metric with other comparison algorithms used is quite significant.

## update after rebuttal
After discussing with the authors, I believe the research problem addressed in this paper is important and the methodology employed demonstrates novelty. However, some aspects of the presentation remain unclear. Taking these factors into consideration, I have decided to raise my score to 3.

**Claims And Evidence:**

I raise doubts about the label distribution assumption proposed in the paper. The authors believe that in the existing label distribution learning, the labels in the label space are insufficient to describe the samples, which contradicts the basic assumption in the consensus definition of label distribution learning that ‘the labels in the label space can fully describe each sample.’ The approach taken in this paper is to add an extra ‘concentration’ to each label distribution component, meaning that each label distribution component in the dataset requires an additional quantity to correctly describe each sample. Therefore, in my opinion, the issue raised by this paper points to the difference between the relative description level of label distribution description and the absolute description level of the original description.

**Essential References Not Discussed:**

After my review, I believe that the paper has adequately cited and discussed the prior work related to its key contributions. I have not identified any uncited or undiscussed work that is essential for understanding the context or the key contributions of the paper.

**Experimental Designs Or Analyses:**

Yes, I suspect that it is not sufficient to test the effectiveness of the algorithm solely through the Friedman test on the average ranking, as this is influenced by the number of comparison algorithms and the quantity of datasets. Would it be better to add a two-tailed t-test to verify the superiority or inferiority against each comparison algorithm?

**Methods And Evaluation Criteria:**

I am rather skeptical about the significance of comparing the experiments in this paper, which involve label distribution learning with concentration terms, to traditional label distribution datasets that do not include concentration terms.

**Other Comments Or Suggestions:**

The authors should use appropriate examples in the Introduction section to match the methods of solving the problem with the issues and motivations they propose. The example of emotional intensity used by the authors is a good illustration, as it appropriately reveals that the same label distribution may correspond to completely different levels of emotional intensity. The examples used in the Introduction can also be resolved by adding labels and other means.
The authors should also be mindful of the perspective from which they articulate the problem, ensuring that it does not lead to ambiguity for the readers. They should avoid giving the impression that “the issue pointed out by the authors is that the existing labels in the label space of label distribution learning cannot fully describe the samples,” as this would contradict the fundamental assumption of label distribution learning.

**Other Strengths And Weaknesses:**

Strengths:
1. The issue addressed in the paper is indeed one of the problems in label distribution learning, namely, the inconsistency in the intensity of the original label descriptions and the label distribution descriptions after normalization. The authors introduce a new concept of concentration in label distribution to address the mismatch in descriptive strength between the relative description level of label distribution and the absolute description level of labels.
2. The authors propose a new framework for label distribution learning based on the concept of concentration, offering new methods and perspectives for label distribution learning.

Weaknesses:
1. The real-world examples used by the authors to illustrate the role of concentration are inappropriate; both sets of examples could resolve the mismatch in descriptive strength by adding extra labels, which is not convincing.
2. The validation of the algorithm’s performance should not be considered as one of the contributions of the entire paper.
3. No theoretical analysis or claims are provided to further validate the robustness and effectiveness of the proposal.

**Questions For Authors:**

In the example used by the authors in the motivation part, such as the example about the boat, if an additional labels, such as “mountain” or “water,” is added to make the label set meet the condition of “being able to fully describe each sample,” does the issue discussed by the authors no longer exist in the example they used?

**Relation To Broader Scientific Literature:**

This paper points out a new issue related to label distribution learning, which has not been considered before.

**Theoretical Claims:**

This paper does not provide any proofs or theoretical analysis.

---

> ### Author Rebuttal · Authors · 2025-04-01
>
> Thank you for your precious suggestions. After careful consideration, our responses to the **questions** you mentioned are listed as follows.
>
> **Claims And Evidence: The authors believe that in the existing label distribution learning, the labels in the label space are insufficient to describe the samples, which contradicts the basic assumption in the consensus definition of label distribution learning.
>
> The assumption that "the labels in the label space can fully describe each sample" is correct in the cases of label distribution learning, which aims to learn the relative description level of each sample. However, in concentration distribution learning, we focus on learning the absolute description level. In these cases, the labels in the label space are insufficient to describe the samples because they overlook the existence of background concentration. In other words, CDL is an extension of LDL, which are not contradicted with each other.
>
> **Methods And Evaluation Criteria: I am rather skeptical about the significance of comparing the experiments in this paper, which involve label distribution learning with concentration terms, to traditional label distribution datasets that do not include concentration terms.**
>
> The paradigm of concentration distribution learning is a new concept, so there are no available CDL datasets. To this end, we construct the first real-world CDL dataset in our paper and conduct experiments on it to verify the effectiveness of our model of exvacating background concentration. And by hiding the last dimension of LDL datasets, we transform LDL datasets into CDL datasets. If our model can recover this hidden dimension, which is regarded as the background concentration, its ability to learn concentration distribution can be proved.
>
> **Experimental Designs Or Analyses: Would it be better to add a two-tailed t-test to verify the superiority or inferiority against each comparison algorithm?**
>
> Thank you for pointing out our shortcoming. The following is the result of the two-tailed t-test. In the significance level of $\alpha=0.01$, the null hypotheses that "there is no significant difference between the ranks of our proposed CDL-LD and the baseline algorithm" are all rejected, which means that the control model CDL-LD is significantly different to all the other baseline algorithms in rank. This result further proves the effectiveness of the proposed model.
>
> | p-value | LDLLC     | SA-IIS    | LCLR      | LDLFs     | LDLLDM    | DLDL      |
> |---------|-----------|-----------|-----------|-----------|-----------|-----------|
> | CDL-LD  | 8.5504e-6 | 2.2121e-4 | 2.9684e-4 | 1.6835e-4 | 1.4473e-4 | 8.7229e-5 |
>
> **Weakness 1: The real-world examples used by the authors to illustrate the role of concentration are inappropriate; both sets of examples could resolve the mismatch in descriptive strength by adding extra labels, which is not convincing.**
>
> The concept of background concentration cannot be replaced by simply adding extra labels. In Fig. 1(a) and (b), background concentration can be interpreted as the background of the image. However, it's hard to describe the background by adding labels like “mountain” or “water” since there are lots of contents in the background except mountain and water. Even if we make out all objects in the background, the label distribution of them is still costly and time-consuming to obtain. For other cases of Fig. 1(c), (d) and Fig.2, the meaning of background concentration is too abstract that it's impossible to be labeled. So we have no choice but to introduce the concept of background concentration.
>
> **Weakness 2: The validation of the algorithm’s performance should not be considered as one of the contributions of the entire paper.**
>
> Our contribution is mainly reflected in coming up with the concentration distribution learning paradigm, constructing the first real-world CDL dataset and proposing a model to learn CDs from LDL datasets. We will rearrange the part of contribution in the final version of this paper.
>
> **Weakness 3: No theoretical analysis or claims are provided to further validate the robustness and effectiveness of the proposal.**
>
> Please refer to the response of **Weakness 1** to reviewer 3FJ2.
>
> **Other Comments Or Suggestions: The authors should use appropriate examples to match the methods of solving the problem with the issues and motivations they propose, and they should avoid giving the impression that contradicts the fundamental assumption of label distribution learning.**
>
> Please refer to **Claims And Evidence** and **Weakness 1** of this response.
>
> **Questions For Authors: If an additional labels, such as “mountain” or “water,” is added to make the label set meet the condition of “being able to fully describe each sample,” does the issue discussed by the authors no longer exist in the example they used?**
>
> Please refer to **Weakness 1** of this response.

---

> > ### Comment · Reviewer_5CcH · 2025-04-02
> >
> > I am glad to see the authors' reply. Firstly, the research motivation of this paper is that the existing LDL framework can only answer questions about relative description degree, but not absolute description degree. On this point, we have reached a consensus, and this work is also very important and interesting. Secondly, for the definition and characterization of the concept of absolute description degree, I think this is where our differences lie. This paper gives the concept of background concentration and explains it with examples, but in my opinion, it is easy to regard absolute description degree as the proportion of each label in the entire image, and therefore, it is easy to solve by simply adding a “background label”. This solution conflicts with the original assumption of LDL. The authors replied that this background is not simply adding a label (or multiple labels), which I semantically agree with, but in terms of expression and subsequent processing, it can be equivalent to adding a label, just called “background”. Thirdly, for the concept of absolute description degree, especially for some non-image data, I currently have no good definition. I appreciate the authors' attempt on this issue, but this attempt or some descriptions in this paper have not yet been well aligned with the LDL learning framework (because this paper is an extension of LDL, so the basic assumptions of LDL need to be considered), which is my main concern. I look forward to seeing further replies from the authors to discuss this issue, and I will further modify my score based on the discussion results.

---

> > > ### Author Response · Authors · 2025-04-04
> > >
> > > Thank you for actively participating in the discussion and your valuable comments. After careful consideration, we will address your concerns point by point.
> > >
> > > First, we agree that the basic label distribution learning assumption that “the labels in the label space can fully describe each sample” is correct. Still, it holds only under the ideal circumstances that the complete label space of instances can be found properly. In fact, most existing LDL datasets do not conform to this property because it’s nearly impossible to annotate all the possible labels of an instance in the real world. As you mentioned, we could add a “background label” to extend the existing LDL datasets. But, to the best of our knowledge, **no existing LDL datasets have been annotated with a label called “background”**. To this end, estimating the value of the “background label” for an existing LDL dataset becomes an important and interesting problem, **which has not been investigated**. To solve this problem, we propose a new framework called concentration distribution learning (CDL) that can estimate the “background label” for an existing LDL dataset without a “background label”.
> > >
> > > Then, you mentioned that “in terms of expression and subsequent processing, it can be equivalent to adding a label called background”, which is correct. Since we want to study the background concentration, **we must assign a value** to the background concentration in mathematical expression and processing, which is formally equivalent to adding a label called "background". Nevertheless, we need to clarify that **the value of background concentration is far more general than the description of the label "background"**. In the image examples in Figs. 1 (a) and (b), we could add a label called "background", and in this case, the background concentration value roughly equals the “background label”. However, in the material composition estimation example in Figs. 1 (c) and (d), there is no material called "background". A more striking example (emotion description) is shown in Fig. 2; emotion intensity is obviously not related to "background". So the proposed concentration is a much more general concept than “background label”, which can be well captured by the proposed CDL framework.
> > >
> > > Furthermore, we want to explain the relevance and difference between LDL and the proposed CDL. LDL illustrates how important the **visible labels** are to corresponding instances. However, the complete label space consists of visible labels and **invisible labels** because it‘s impossible to annotate all the possible labels of an instance. The existing LDL methods only focus on the relative importance among the visible labels, and overlook the proportion of the invisible ones. We find out that **ignoring invisible labels can sometimes lead to confusion**. Hence, it’s also necessary to learn their description degree, and the paradigm of concentration distribution learning (CDL) is proposed naturally. The invisible labels may have a clear actual meaning, such as the real background of an image. But in some cases, they can also be **very abstract**, like the description degree of “no emotion” on the human face dataset, so we use “background concentration” as a **universal designation** for the description degree of invisible labels. It's actually a very general concept. By learning an additional background concentration term, CDL addresses the limitation of LDL properly and expands the research depth of LDL.
> > >
> > > Finally, thank you again for your valuable comments. In the final version of our paper, we will modify the introduction to describe the CDL framework clearly and avoid creating ambiguity for the readers.

---

### Official Review · Reviewer_7WaC · 2025-03-14

**Overall Recommendation:** 3

**Summary:**

The paper introduces a novel concept called concentration distribution learning (CDL), which extends traditional label distribution learning (LDL) by incorporating a background concentration term. This term represents the absolute description degree of labels not present in the existing label space. The authors propose a method called CDLLD that uses probabilistic methods and neural networks to learn both label distributions and background concentrations from existing LDL datasets. The paper demonstrates the effectiveness of this approach through extensive experiments on multiple real-world datasets demonstrate the effectiveness of the proposed method. ## update after rebuttal. Most of my concerns are addressed, the authors have conducted experiments on larger datasets and clarified the unclear parts, thus I have decided to retain my positive score.

**Claims And Evidence:**

The authors claim that "excavating the background concentration makes full use of the information in the datasets and benefits the downstream tasks", however, the experiments are not conducted on several downstream tasks, I suggest the authors should be cautious when making such a statement.

**Essential References Not Discussed:**

Yes, there is one essential reference that should be discussed.

[1] "Label distribution changing learning with sample space expanding." Journal of Machine Learning Research 24.36 (2023): 1-48.

**Experimental Designs Or Analyses:**

Yes, I have checked the experimental parts, the experiments in this paper are useful for validating the effectiveness of the proposed method.

**Methods And Evaluation Criteria:**

Yes, the proposed method and evaluation criteria in this paper are well-suited for the problem of label distribution learning.

**Other Comments Or Suggestions:**

Some parts need further explanation. See the question parts below.

**Other Strengths And Weaknesses:**

Strengths:
1. The problem studied in this paper is interesting and important in the label distribution learning area.
2. The proposed method is effective according to the experimental results.
3. The authors construct the first real-world concentration distribution dataset.

Weaknesses:
1. Some parts of the paper are not easy to understand and need further explanation. See the question parts below.
2. Datasets used in the experiments are small, larger datasets should be considered.

**Questions For Authors:**

1. Why do you choose the Dirichlet distribution(line 149 ), please give the reasons behind.
2. "In Eq. (1), assuming that the background concentration is evenly spread on each class of the real label distribution vector b in probability." Is this assumption reasonable? Maybe it is better to consider uneven spread.
3. Why define Eq.(10) in such a form? Can you make a detailed explanation?

**Relation To Broader Scientific Literature:**

The key contributions of the tempered sigmoid proposed in this paper are closely related to several areas of the broader scientific literature, particularly in the fields of label distribution learning, facial expression recognition, and emotion recognition.

**Theoretical Claims:**

No theoretical claim is provided in this paper.

---

> ### Author Rebuttal · Authors · 2025-04-01
>
> Thanks for your valuable reviews, and your suggestions will effectively help us improve our work. Below are our responses to your **questions**.
>
> **Claims And Evidence: The authors claim that "excavating the background concentration makes full use of the information in the datasets and benefits the downstream tasks", however, the experiments are not conducted on several downstream tasks, I suggest the authors should be cautious when making such a statement.**
>
> Thank you for your advice. We think that in the experiments on the real-world CDL dataset, our model exvacates the hidden strength of emotions and helps us distinguish the images better, which can also be regarded as a downstream task. So we drew the conclusion that our proposed model can benefit the downstream tasks, which is reasonable.
>
> **Essential References Not Discussed: There is one essential reference that should be discussed, that is [1] "Label distribution changing learning with sample space expanding." Journal of Machine Learning Research 24.36 (2023): 1-48.**
>
> Thanks for pointing out our missing reference. We will add this to the final version of this paper.
>
> **Theoretical Claims: No theoretical claim is provided in this paper.**
>
> Please refer to the response of **Weakness 1** to reviewer 3FJ2.
>
> **Weakness 2: Datasets used in the experiments are small, larger datasets should be considered.**
>
> Thank you for your advice. We carry out further experiments on the human gene dataset (with 17,892 instances). The results are listed as below, with the best result of each metric shown in boldface. The extensive experiment further proves the superiority of the proposed CDL-LD.
>
> |        |      Cheby↓      |      Clark↓      |       LCLR       |      Cosine↑     |
> |:------:|:----------------:|:----------------:|:----------------:|:----------------:|
> | CDL-LD | **0.5735±.0090** | **3.6863±.0288** | **1.0830±.0160** | **0.7757±.0025** |
> |  LDLLC |   0.6745±.0039   |   3.9021±.0226   |   2.2932±.0206   |   0.6199±.0179   |
> | SA-IIS |   0.6948±.0121   |   3.7957±.0359   |   2.1820±.0337   |   0.6114±.0483   |
> |  LCLR  |   0.6148±.0178   |   3.5548±.0387   |   2.3030±.0228   |   0.6514±.0150   |
> |  LDLFs |   0.6383±.0467   |   3.7784±.0700   |   2.7793±.0261   |   0.6248±.0140   |
> | LDLLDM |   0.6549±.0283   |   3.8324±.0307   |   2.4558±.0193   |   0.6719±.0248   |
> |  DLDL  |   0.6056±.0259   |   3.7024±.0076   |   2.1174±.0328   |   0.6854±.0144   |
>
> **Question 1: Why do you choose the Dirichlet distribution(line 149 ), please give the reasons behind.**
>
> Dirichlet distribution is a probability distribution on a multidimensional space and the sum of its components is 1, which makes it suitable for representing the probability distribution of multiple mutually exclusive events. Label distributions represent the probability distribution of multiple labels which are mutually exclusive, so we choose Dirichlet distribution in this paper.
>
> **Question 2: In Eq. (1), assuming that the background concentration is evenly spread on each class of the real label distribution vector b in probability." Is this assumption reasonable?**
>
> The paradigm of concentration distribution learning is a new concept raised by this paper. Since we have no further prior knowledge about the background concentration distribution pattern, the assumption that it is evenly spread across each class of the real label distribution is general. Furthermore, the experimental results also validate the rationality of this assumption.
>
> **Question 3: Why define Eq.(10) in such a form? Can you make a detailed explanation?**
>
> $\vert\vert\boldsymbol{y}-\boldsymbol{p}\vert\vert_2^2$ is the MSE loss mentioned above in the paper, and $\frac{1}{B(\boldsymbol{\alpha})}\prod^c_{i=1}p_i^{\alpha_i-1}$ is the probability density function of Dirichlet distribution. Integrating the product of these two terms gives the final loss function of the neural network, which aims to minimize the average value of MSE loss over the whole Dirichlet distribution.

---

### Official Review · Reviewer_3FJ2 · 2025-03-16

**Overall Recommendation:** 3

**Summary:**

This paper proposes Concentration Distribution Learning (CDL), which introduces background concentration to address the limitation of Label Distribution Learning (LDL) in capturing hidden information. The authors designed the CDL-LD model based on the Dirichlet distribution, combining confidence and background concentration to generate concentration distributions. They also constructed the SJA c dataset to validate the approach. Experiments demonstrate that CDL-LD outperforms existing methods across multiple metrics, especially in concentration distribution tasks.

**Claims And Evidence:**

1.Why does the paper assume that each alpha_i is composed of the sum of the dataset e_i and the background concentration u_i, instead of treating the background as a whole? Specifically, why not define αi = e_i, with the sum of alpha_i plus a single overall u equaling 1? The authors' method seems inconsistent with the idea presented in the introduction.
2.Why does the paper assume that the background concentration is evenly spread across each class of the real label distribution? There is neither theoretical nor intuitive justification for this assumption.

**Essential References Not Discussed:**

More comprehensive discussion over related works from the last 2-3 years should be incorporated.

**Experimental Designs Or Analyses:**

The paper provides a detailed introduction to the experimental design and analysis, but the datasets used seem relatively small. Have you tried testing on larger datasets?

**Methods And Evaluation Criteria:**

The paper uses MSE loss in a way that seems somewhat unreasonable. Have you tried using CE loss instead?
The idea in the paper is relatively novel, the method is somewhat simplistic.

**Other Comments Or Suggestions:**

N/A

**Other Strengths And Weaknesses:**

Advantages:
The paper presents a novel and relatively reasonable approach.
The introduction to related work and the explanation of the experimental section are quite detailed.

Disadvantages:
Some of the theoretical assumptions in the paper lack proper analysis or proof.
The datasets used in the experimental section are relatively small, and no experiments have been conducted on larger datasets.

**Questions For Authors:**

The authors should carefully address the above-mentioned issues.

**Relation To Broader Scientific Literature:**

The paper is related to label distribution learning, concentration distribution learning,  and face expression recogintion.

**Theoretical Claims:**

N/A

---

> ### Author Rebuttal · Authors · 2025-04-01
>
> Thanks for your valuable reviews, and your suggestions will effectively help us improve our work. Below are our responses to your **questions**.
>
> **Claims And Evidence 1 \& 2: Why does the paper assume that each $alpha\_i$ is composed of the sum of the dataset $e\_i$ and the background concentration $u\_i$, instead of treating the background as a whole?**
>
> The paradigm of concentration distribution learning is a new concept raised by this paper. Since we have no further prior knowledge about the background concentration distribution pattern, the assumption that it is evenly spread across each class of the real label distribution is general. Furthermore, the experimental results also validate the rationality of this assumption. By defining $\alpha_i=e_i+u_i$, we make the problem mathematically computable and finally draw the conclusion that all $u_i$ equal to one, which indicates that we should treat background concentrations as a whole. In other words, without initially dividing the background concentrations, the value of $u$ cannot be decided.
>
> **Methods And Evaluation Criteria 1: The paper uses MSE loss in a way that seems somewhat unreasonable. Have you tried using CE loss instead?**
>
> By applying MSE loss, the final loss function can be arranged in the form of Eq. (10) in the paper, which minimizes the predictive square error ($(y_i-\frac{\alpha_i}{S})^2$) and the variance of the Dirichlet distribution ($\frac{\alpha_i(S-\alpha_i)}{S^2(S+1)}$) simultaneously. Optimizing these two terms makes our experimental results more precise and stable. The CE loss, however, does not have this property.
>
> **Relation To Broader Scientific Literature: The paper studies label distribution learning, but only ancient references are cited.**
>
> Thank you for your pointing out our shortcoming. We will incorporate some recent works in the final version of our paper.
>
> **Disadvantage 1: Some of the theoretical assumptions in the paper lack proper analysis or proof.**
>
> Thank you for pointing out our shorcoming. We apply rademacher complexity to explain the theoretical bound of our proposed model.
>
> Let $\mathcal{H}$ be a family of functions. For any $\delta>0$, with probability at least $1-\delta$, for all $h \in \mathcal{H}$ such that
>
> $$
>  \mathcal{L}(h) \leq \mathcal{L}\_{S}(h)+\widehat{\mathcal{R}}\_{S}(\mathcal{H})+3\sqrt{\frac{\log 2 / \delta}{2 n}},
> $$
>
> where $\mathcal{L}(h)$ and $\mathcal{L}\_{S}(h)$ are the generalization risk and empirical risk with respective to h, and $\widehat{\mathcal{R}}\_{S}$ is the empirical rademacher complexity bounded by $(\mathcal{H})\leq\mathbb{E}\_{\boldsymbol{\sigma}}\left[ \frac{1}{n} \sum\_{i=1}^{n} \sigma\_{i} \mathcal{L}\_{AMSE}(\mathbf{\alpha}\_i) \right]$ with $\sigma\_{i}\in[0,1]$. $n$ represents the number of instances.
>
> Because $\mathcal{L}\_{AMSE}(\mathbf{\alpha})=\sum^c\_{i=1}(y\_i-\frac{\alpha\_i}{S})^2+\frac{\alpha\_i(S-\alpha\_i)}{S^2(S+1)}>0$, we get
>
> $$
>         \mathcal{L}(h)- \mathcal{L}\_{S}(h) \leq \mathbb{E}\_{\boldsymbol{\sigma}}\left[ \frac{1}{n} \sum\_{i=1}^{n} \sigma\_{i} \mathcal{L}\_{AMSE}(\mathbf{\alpha}\_i) \right]+3\sqrt{\frac{\log 2 / \delta}{2 n}}
>         \leq \frac{1}{n} \sum\_{i=1}^{n} \sum^c\_{j=1} \left[(y\_{ij}-\frac{\alpha\_{ij}}{S\_i})^2+\frac{\alpha\_{ij}(S\_i-\alpha\_{ij})}{S\_i^2(S\_i+1)}\right]+3\sqrt{\frac{\log 2 / \delta}{2 n}},
> $$
>
> in which $c$ is the number of classes. We assume that the neural network gives $e>0$ for every instance, then $\forall i,j, \alpha\_{ij}>1, S\_i>c$. So we have
>
> $$
>         \mathcal{L}(h)- \mathcal{L}\_{S}(h) \leq  \frac{1}{n} \sum\_{i=1}^{n} \sum^c\_{j=1} \left[(y\_{ij}-\frac{\alpha\_{ij}}{S\_i})^2+\frac{\frac{\alpha\_{ij}}{S\_i}(1-\frac{\alpha\_{ij}}{S\_i})}{S\_i(S\_i+1)}\right]+3\sqrt{\frac{\log 2 / \delta}{2 n}}
>         \leq \frac{1}{n} \sum\_{i=1}^{n}  \left[1+\frac{1}{4c(c+1)}\right]+3\sqrt{\frac{\log 2 / \delta}{2 n}}.
> $$
>
> When the number of instances $n$ tends to infinity, the bound finally becomes $1+\frac{1}{4c(c+1)}$, which indicates that this bound shrinks when the number of classes $c$ increases. **This conclusion is intuitive because the background concentration tends to zero when $c$ tends to infinity, degrading the CDL problem to learnable LDL problem.**
>
> **Experimental Designs Or Analyses \& Disadvantage 3: The paper provides a detailed introduction to the experimental design and analysis, but the datasets used seem relatively small. Have you tried testing on larger datasets? \& The datasets used in the experimental section are relatively small, and no experiments have been conducted on larger datasets.**
>
> Thank you for your advice. We carry out further experiments on the human gene dataset (with 17,892 instances). The results are listed as in the response of **Weakness 2** to Reviewer 7WaC due to the character limit, which further proves the superiority of the proposed CDL-LD.

---

### Decision · Program_Chairs · 2025-05-01

**Decision:**

Accept (poster)

**Comment:**

This paper introduces a novel technique for label distribution learning, specifically concentration distribution learning, which incorporates a background concentration term. The experimental results demonstrate that the proposed method is effective on existing benchmark datasets. During the rebuttal phase, the authors addressed most concerns raised by the reviewers, and all reviewers now support the acceptance of this paper. Therefore, I recommend accepting this paper. The authors should revise the manuscript according to the reviewers' feedback.